# Development of a High Performance Liquid Chromatography with Diode Array Detector (HPLC-DAD) Method for Determination of Biogenic Amines in Ripened Cheeses

**DOI:** 10.3390/molecules27238194

**Published:** 2022-11-24

**Authors:** Marzena Pawul-Gruba, Tomasz Kiljanek, Anna Madejska, Jacek Osek

**Affiliations:** 1Department of Hygiene of Food of Animal Origin, National Veterinary Research Institute, Partyzantów 57 Avenue, 24-100 Pulawy, Poland; 2Department of Pharmacology and Toxicology, National Veterinary Research Institute, Partyzantów 57 Avenue, 24-100 Pulawy, Poland

**Keywords:** biogenic amines, cheese, pre-column derivatization, HPLC

## Abstract

Biogenic amines (BAs) are organic, basic nitrogenous compounds formed during the decarboxylation of amino acids. A method for the determination of eight biogenic amines (tryptamine, 2-phenyletylamine, putrescine, cadaverine, histamine, tyramine, spermidine, spermine) in ripened cheeses was developed and validated. Cheese samples with the addition of internal standards were extracted with 0.2 M perchloric acid and pre-column derivatized with dansyl chloride at 60 °C for 15 min, purified with toluene and dried under a stream of nitrogen. The samples were analyzed using high performance liquid chromatography with diode array detector (HPLC-DAD). The method was validated with the BAs at three concentration levels: 50, 100, and 200 mg/kg, respectively. The obtained values of correlation coefficient (R^2^) ranged at 0.9997–0.9998 for all of compounds. The limits of detection (LOD) and quantification (LOQ) were in ranges 1.53–1.88 and 5.13–6.28 mg/kg, respectively. The recovery for all of biogenic amines ranged from 70 to 120% and the precision (RSDr) value were <20%. The validated method was applied to analysis of 35 real ripened cheese samples purchased in Poland.

## 1. Introduction

Biogenic amines (BAs) are organic, basic nitrogenous compounds with aliphatic (putrescine, cadaverine, spermine, spermidine), aromatic (tyramine, 2-phenylethylamine), or heterocyclic (histamine, tryptamine) structures [1]. They are formed during the decarboxylation of amino acids or amination and the transamination of aldehydes and ketones [2,3].

Biogenic amines are widespread in the environment, they occur naturally in animals, plants, and bacterial cells. BAs are responsible for biological processes such as synaptic transmission, blood pressure control, allergic reaction, and cell growth control [2]. However, biogenic amines can also be dangerous for human health, if their level in food or drink is high. Foods that may contain high levels of biogenic amines include fish, and other food products, which technology relies on for fermentation (cheese, meat, vegetables, beer, and wine) [4,5]. 

BAs ingested with food are detoxified in the digestive tract, which is able to metabolize their low amounts by monoamine and diamine oxidase enzymes [6,7]. However, upon the intake of higher levels of biogenic amines, the natural detoxification system is unable to eliminate them sufficiently [8,9]. The BAs that cause the most common and serious poisoning are histamine and tyramine. The symptoms of intoxication are urticaria, hypotension, palpitation, diarrhea, headache, nausea, and vomiting. Furthermore, 2-phenylethylamine can be a migraine inductor [6,10,11,12]. Putrescine and cadaverine can potentiate the toxicity of tyramine and histamine and may also react with nitrites to form carcinogenic N-nitroso compounds. Putrescine, in turn, may also affect the abnormal cells transformation [2,5,13]. 

The factors influencing the formation of biogenic amines in food are the availability of free amino acids, the presence of bacteria that release amino acid decarboxylases, and favorable conditions for their growth and activity [14,15]. The amount and type of BAs formed in food are highly influenced by internal factors such as pH, water activity, composition and level of bacterial microflora, and external conditions such as time and temperature, which allow bacteria to grow during food processing and storage [13,16].

Although ripened cheeses are one of the most important categories of food in which significant amounts of biogenic amines may be present, there are no regulatory criteria for their maximum acceptable level. The European Commission in Regulation 2073/2005 specifies only the maximum level of histamine in fish and fish products [17]. The high content of protein and free amino acids in ripened cheeses create favorable conditions for the growth of bacteria that can produce toxic biogenic amines. Their formation is also highly influenced by temperature and humidity during the maturation of this food category [1,2,18]. 

For the analysis of biogenic amines, various techniques are used, as thin-layer chromatography (TLC), gas chromatography (GC), high performance liquid chromatography (HPLC), the capillary electrophoretic method (CE), and biochemical assays [5,19]. The most appropriate technique for the determination of BAs in food is HPLC with reversed-phase separation [20]. HPLC methods are more preferable than GC because biogenic amines are low volatile and have the functional groups to be derivatized for spectrophotometric detections. The application of derivatization in chromatography is the subject of many studies [21,22,23]. This process consists of converting the analytes into appropriate derivatives with properties enabling their chromatographic separation. Derivatization is a very helpful technique for the determination of analytes in complex matrices such as foods. Therefore, most methods require that biogenic amines should be derivatized before detection because they do not have enough absorption in the UV-Vis or fluorescence wavelength ranges [5,19]. Derivatization agents that are usually used to reach the levels of selectivity and sensitivity required for the BAs determination are dansyl chloride, dabsyl chloride, benzoyl chloride, fluorescein isothiocyanate, phenyl isothiocyanate, 9-fluorenyl methyl chloroformate, fluorescamine, AQC (6-aminoquinolyl-N-hydroxysuccinimidyl carbamate), and o-phthalaldehyde as a post-column derivatization reagent [5,24,25].

The aim of the present study was to develop and validate a method for determination of eight biogenic amines (tryptamine, 2-phenylethylamine, putrescine, cadaverine, histamine, tyramine, spermidine and spermine) in ripened cheeses using a high performance liquid chromatography with diode array detector (HPLC-DAD). This method was then applied to analyze the presence and level of BAs in samples of commercially available ripened cheeses.

## 2. Results and Discussion

### 2.1. Optimization of Extraction Step

The extraction of biogenic amines from ripened cheeses is a difficult process due to a large amount of milk fat in the matrix. In order to optimize this step, the suitability of various solvents for the extraction of BAs from ripened cheeses was investigated. The following solutions were tested: perchloric acid (HClO_4_; 0.2 M, 0.4 M, 0.6 M), hydrochloric acid (HCl; 0.1 M) and trichloroacetic acid (TCA; 5%, 10%), respectively. Camembert cheese spiked with 100 mg/kg of each BAs was used. Each sample was analyzed three times and then the recoveries for all amines were calculated (Figure 1). The best results were obtained when perchloric acid at concentration of 0.2 M as the extraction solvent was applied. 

Various solvents were used for the extraction of Bas, taking into consideration the characteristics of the analyzed samples. Solutions of 0.2 M perchloric acid was previously proposed for fish products [26], 0.4 M and 0.6 M for cheeses [5,27]. Hydrochloric acid 0.1 M was also applied before for cheeses [20], 5% trichloroacetic acid for meat [28], and 10% TCA for various foods [29]. 

In the present study, the highest extraction efficiency was obtained with 0.2 M perchloric acid. Therefore, this solvent, originally described in the methodology for the determination of BAs in fish [26], was used for the extraction of the amines from cheeses.

According to the literature, some authors used double [20] or triple [5] extraction to determine the content of biogenic amines; therefore, the number of extractions was also currently optimized. For this purpose, samples of Camembert cheese were spiked with 100 mg/kg of each BAs, 100 µL of the internal standard (5 mg/mL) and were subjected to single and double extraction with 0.2 M perchloric acid. The double extraction process consisted of collecting the supernatant from the first extraction and the residue was extracted again with 10 mL of 0.2 M HClO_4_. These two aqueous extracts were combined and adjusted to 25 mL with 0.2 M perchloric acid. The results obtained during single and double extractions are shown in Table 1. Double extraction has not significantly improved the recovery of the biogenic amines tested. The recoveries were very similar to those obtained in single extraction but more contaminant peaks were visible (data not shown). Furthermore, this sample preparation process required more time than the single extracted step. Initially, when using calibration curves based on peak areas, low recoveries ranging from 50–60% were obtained for some biogenic amines. Therefore, a calibration with an internal standard was used, which allowed to obtain recoveries of over 70% for all analytes.

### 2.2. Optimization of Derivatization Step

In the present study, four reaction conditions for derivatization with dansyl chloride (incubation at 25 °C for 15 min, at 40 °C for 45 min, at 60 °C for 5 min, and at 60 °C for 15 min) were tested using a standard solution of biogenic amines at a concentration of 25 µg/mL. For each of the four derivatization conditions, standard solutions were analyzed three times and then the mean values of the peak areas for all amines were calculated. It was shown that derivatization at 60 °C for 15 min had the best effect on the peak areas size of BAs (Table 2). 

Different authors reported various derivatization conditions of biogenic amines with dansyl chloride, and the process performed at 40 °C for 45 min was mostly used [30,31]. Other authors published the derivatization conditions at 60 °C for 5 min, or at 25 °C for 15 min [26,32]. 

The results of our study showed that derivatization at higher temperatures resulted in greater peak areas of standard solutions, compared to incubation at lower temperatures, which is consistent with another study [20].

### 2.3. Method Validation

Standard solutions of eight biogenic amines in the range from 5 to 200 mg/kg of each, containing 1,7-diaminoheptane as an internal standard, were used for calibration after their derivatization. The limit of detection (LOD) and limit of quantification (LOQ) were in the ranges of 1.53–1.88 and 5.13–6.28 mg/kg, respectively. The developed method demonstrated a good linearity for all BAs in the range of LOQ-200 mg/kg. The obtained values of correlation coefficient (R^2^) were within the range of 0.9997–0.9998 for all of compounds (Table 3).

The following types of ripened cheeses were used for validation: Camembert, Brie, Saint Agur, Blu Rival, Limburgish, and Grana Padano. For the evaluation of precision and recovery, blank cheese samples were spiked at three BAs concentration levels of 50, 100, and 200 mg/kg, respectively. The results concerning recovery and precision are presented in Table 4. The relative standard deviations (RSDs) for the intra-day precision ranged from 0.7% (tyramine) to 3.0% (putrescine), whereas for the inter-day precision were between 4.1% (cadaverine) and 13.5% (spermine). These results confirmed the good repeatability and intermediate precision of the developed method. The recovery values obtained for all biogenic amines were within the satisfactory range of 70–120%. Thus, the validation results showed that the method developed was suitable for determination of biogenic amines in cheese. 

The method performance was verified by analysis of the reference material (canned fish TYG018RM, histamine reference value: 220 ± 5 mg/kg, FAPAS, York, England), and the obtained result was 219.8 mg/kg.

### 2.4. Real Samples Application

A total of 35 different types of cheeses were analyzed. It was shown that 30 (85.7%) samples contained at least one biogenic amine in concentration above LOQ. One or two biogenic amines were detected in 21 samples, three BAs were found in four samples and four amines in three samples. A maximum of five BAs were found in two cheese samples. The results of BAs analysis in ripened cheeses and statistical evaluation are shown in Table 5 and Table 6, respectively. Figure 2 presents the chromatograms: standard solution of eight biogenic amines (25 µg/mL of each) (a), cheese sample spiked with BAs at concentration of 100 mg/kg of each (b), and real cheese sample (Gorgonzola) with three biogenic amines detected (c).

The content of biogenic amines varied depending on the type of cheese (mold-ripened, blue-veined, and hard cheese) and its specific variant. Histamine was detected in 9 (25.7%) samples (Table 5). The concentration of this amine ranged from 6.48 to 127 mg/kg. The highest contents of histamine were found in three samples of Gorgonzola (127, 94.2 and 78.4 mg/kg) and Grana Padano (84.7 mg/kg). One of the ripened cheeses contained histamine above 100 mg/kg which, according to the European Union regulation, is the maximum level of this BA for fish and fish products [17]. In the present study it was observed that higher concentrations of histamine occurred mainly in blue-veined (Gorgonzola) and hard cheeses. In another survey on the content of histamine in cheeses, this BA was detected in 51.2% of Spanish samples in concentrations ranging from 5 to 571 mg/kg, and the highest content of histamine, as in the current investigation, was found in ripened hard cheese made mainly from raw sheep’s milk [18]. Other studies reported that 13.8% of the samples contained histamine above 100 mg/kg, whilst the highest histamine contents were determined in hard regional cheeses (1159.7, 820.6, and 397.2 mg/kg) and Gorgonzola (255.3 mg/kg) [5].

In our study, tyramine was detected in 10 (28.6%) samples in the concentration range of 7.28–692 mg/kg. The highest tyramine contents were detected in a soft cheese with chives (692 mg/kg), Roquefort (631 mg/kg), Emmental (474 mg/kg), and Raclette (287 mg/kg) (Table 5). The tyramine content above 100 mg/kg was found in 5 (17.1%) of samples. In a study performed in Austria, this amine was detected mainly in hard cheeses with the maximum concentration of 486.4 mg/kg in Cantal, the French raw milk cheese [5]. In other studies, high concentrations of tyramine were found in mold-ripened cheese (762.75 mg/kg) and semi hard cheese (767.03 mg/kg) [27].

Cadaverine and putrescine were detected in 10 (28.6%) samples, with the concentration ranged from 10.1 to 208 mg/kg and from 6.57 to 170 mg/kg, respectively (Table 5). The highest contents of this BA were found in three samples: a soft cheese with chives (208 mg/kg), Camembert (162 mg/kg), and Raclette (148 mg/kg). The highest putrescine content was detected in the same sample of Raclette (170 mg/kg). The contents of cadaverine and putrescine higher than 100 mg/kg were in the 8.6% and 2.9% of samples, respectively. In other studies, the highest concentration of cadaverine (748.2 mg/kg) and putrescine (523.2 mg/kg) were found in the same sample of acid curd cheese [5]. 

The concentrations of the remaining BAs tested were much lower compared to the other amines and ranged from 6.91 to 42.5 mg/kg (2-phenylethylamine), 6.46 to 14.1 mg/kg (tryptamine), and 6.68 to 15.1 mg/kg (spermidine). Spermine was not detected in any sample. No detectable level of biogenic amines was identified in five (14.3%) samples such as Brie, Saint Albray, Snack a la francaise, and Limburger. 

The results of our preliminary study showed that tyramine was the most commonly detected biogenic amine in high concentrations, which is consistent with other studies [5,27]. Among the ripened cheeses tested, BAs were mainly detected in hard and blue-veined cheeses, whereas in mold-ripened cheeses the amines were identified with a much less frequency. 

## 3. Materials and Methods

### 3.1. Chemicals and Reagents

High purity biogenic amines of analytical standards (98% or higher): tryptamine dihydrochloride, 2-phenylethylamine dihydrochloride, putrescine dihydrochloride, cadaverine hydrochloride, histamine dihydrochloride, tyramine dihydrochloride, spermidine, spermine were purchased from Sigma-Aldrich (St. Luis, MI, USA) and Acros Organics (Geel, Belgium). The internal standard (IS) 1,7-diaminoheptane was supplied by Sigma-Aldrich. HPLC grade acetone and toluene used for reagent preparation and for extraction were purchased from Merck (Darmstadt, Germany) and Avantor Performance Materials Poland S.A. (Gliwice, Poland). Dansyl chloride from Sigma-Aldrich was used as a derivatization reagent. Perchloric acid used for extraction was supplied by Avantor Performance Materials Poland S.A. Sodium carbonate and L-proline were purchased from Sigma-Aldrich. The HPLC grade acetonitrile used as eluent in liquid chromatography was obtained from Merck. Purified through a Mili-Q Plus system water (Merck Millipore; Billerica, MA, USA) was used for mobile phase and for the standards preparation.

### 3.2. Preparation of Standard Solutions

Stock standard solutions (25 mg/mL) were prepared by weighing the appropriate amount of each standard into a 20 mL volumetric flask and dissolving in water. Mixed standard solution I (2.5 mg/mL) was prepared by water dilution of 5 mL of each stock standard solutions in 50 mL volumetric flask. Mixed stock solution II was prepared from mixed stock solution I by dilution (1/10) with water to yield a concentration of 0.25 mg/mL. Working solutions were prepared by dilution of required aliquots of the mixed standard solution I or II. Internal standard solution (5 mg/mL) was prepared by dissolving 250 mg 1,7-diaminoheptane in 50 mL of water. All solutions were stored at 2–9 °C.

### 3.3. Apparatus and Chromatographic Conditions

The instrumental analysis was performed using HPLC-DAD Varian Model 330 Pro Star (Varian, The Netherlands) equipped with ternary pump, autosampler and column thermostat, controlled by the Galaxie Workstation software. Separation was conducted on Unisol C18 150 × 4.6 mm, 3 μm with precolumn 10 × 3 mm, 3 μm (Agela Technologies, Wilmington, DE, USA). The column oven temperature was maintained at 30 °C and the analytes were separated using acetonitrile (mobile phase A) and water (mobile phase B). The flow rate of 1 mL/min was used. Gradient program was made as follows: 60% (mobile phase A) increased to 75% in 6 min and hold 2 min, increased to 95% in 5 min and hold 7 min, decreased to 60% in 1 min and hold with 9 min to column recondition. Total analysis run time was 30 min, and the injection volume was 10 μL.

### 3.4. Sample Preparation

Five grams of homogenized cheese sample was weighted in a 50 mL centrifuge tube and 100 μL of the internal standard (5 mg/mL, 1,7-diaminoheptane) was added. Then, ceramic homogenizer and 10 mL of 0.2 M hydrochloric acid were added and sample was shaken for 1 min and in an Analog Orbital Shaker (Sunlab Instruments, Germany) for 5 min. The obtained cheese slurry was centrifuged at 3000 rpm for 20 min at 4 °C and the upper fat layer was removed. In case of unclear supernatants, they were filtered through a PTFE filter. After that, 100 μL was transferred into a screw cap glass vial and 200 μL of saturated sodium carbonate solution and 400 μL of dansyl chloride acetone solution (7.5 mg/mL) were added. Next, the mixture was mixed using a vortex and incubated in a heating module for 15 min in the dark at 60 ± 1 °C. Further, 100 μL L-proline (10 mg/mL) was added, shaken manually and the vial was stored in the dark for 15 min. In the next step, 500 μL toluene was added, shaken manually, and stored for 20 min at ≤−18 °C in order to freeze the water phase. Then, as much as possible of the upper organic phase was transferred into a new glass vial and evaporated to dryness under nitrogen at 44 °C. The residue was dissolved in 200 μL (standards in 100 μL) acetonitrile. Before injection into HPLC system the solution was filtered through a 0.2 μm syringe PTFE filter. The sample preparation procedure is shown in Figure 3.

### 3.5. Method Validation

The method was validated in terms of its linearity, limit of detection (LOD), limit of quantification (LOQ), precision, and recovery. The calibration curves were obtained by plotting the peak-area ratios of the analyte to the IS versus the concentration. Linearity was evaluated in duplicate at five concentrations of the standard mixtures levels: 5, 10, 20, 100, 200 mg/kg. The LOD and LOQ values for all analytes were calculated based on the data from the appropriate calibration curves. LOD was established as 3.3 times the value of residual standard deviation (Sx/y) divided to the slope of the calibration curve (a). LOQ was established as 10 times the (Sx,y) divided to (a). The cheeses samples free from biogenic amines were used as blank, to spike aliquots for validation studied. To assess the accuracy of the developed method, precision and recovery were evaluated. Spiked of the same cheese sample in six replicates for each level analyzed within one working day were used to estimate the intra-day precision. The inter-day precision was assessed for each level based eighteen replicates spiked of the various cheese samples within three working days. The method’s precision was determined by calculating the relative standard deviation (RSDr). Average recovery percentages were calculated by comparison the determined concentrations of spiked samples to their target level. 

### 3.6. Real Samples

The developed and validated method was applied to analyze biogenic amines in 35 samples of ripened cheeses. The investigation covered mold-ripened soft cheeses (Brie, Camembert types), blue-veined cheeses (Roquefort, Gorgonzola types) and hard cheeses (Grana Padano, Parmesan types). Cheeses were purchased from various local retail markets in the Pulawy region (Poland) and the samples were immediately delivered to the laboratory at refrigerated temperature.

## 4. Conclusions

This developed and validated method allows analysis of biogenic amines in ripened cheeses by HPLC-DAD. Using perchloric acid (0.2 M) for extraction resulted in better recoveries than with perchloric acid (0.4 M, 0.6 M), hydrochloric acid (0.1 M), and trichloroacetic acid (5%, 10%). In addition, the pre-column derivatization with dansyl chloride at 60 °C for 15 min and a single extraction of samples significantly reduced the time and decreased the cost of the analysis. The present method may be applied for analysis of biogenic amines in different ripened cheeses. 

## Figures and Tables

**Figure 1 molecules-27-08194-f001:**
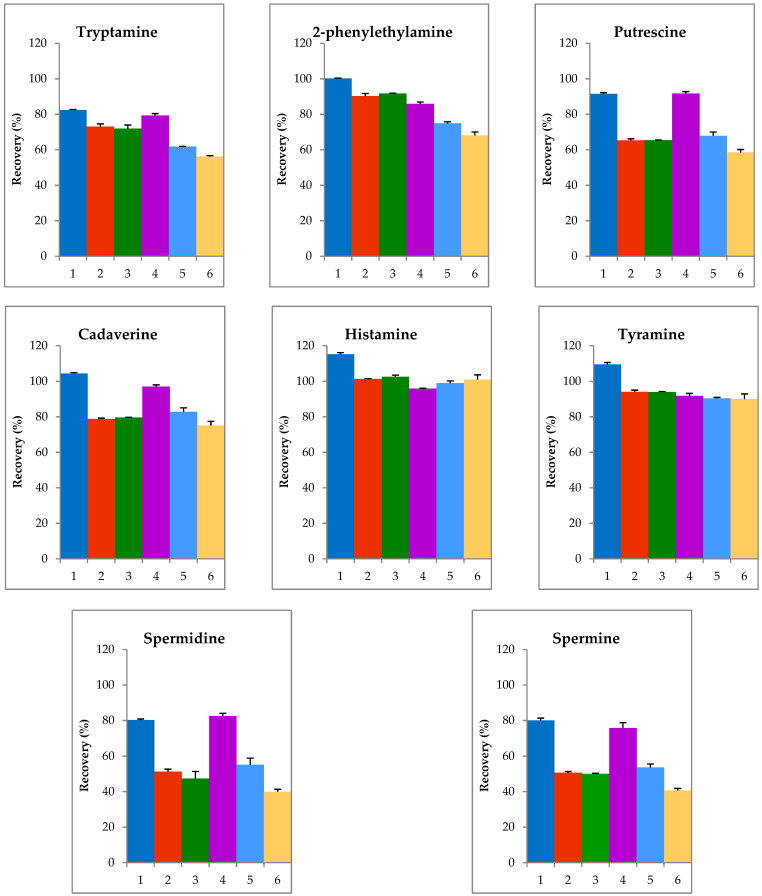
Results of the optimization of the extraction process for biogenic amines using six solutions: 1—0.2 M HClO_4_, 2—0.4 M HClO_4_, 3—0.6 M HClO_4_, 4—0.1 M HCl, 5—5% TCA, and 6—10% TCA.

**Figure 2 molecules-27-08194-f002:**
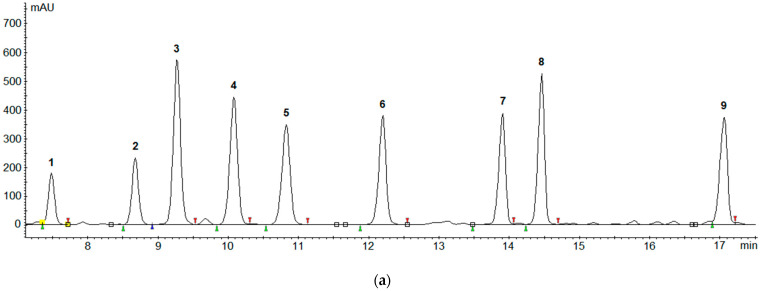
Chromatograms of standard solution with biogenic amines (25 µg/mL) (**a**), cheese sample (Camembert) spiked BAs at concentration of 100 mg/kg (**b**) and real cheese sample (Gorgonzola) with biogenic amines detected (**c**). Identification of the peaks: 1—tryptamine, 2—2-phenylethylamine, 3—putrescine, 4—cadaverine, 5—histamine, 6—1,7-diaminoheptane (IS), 7—tyramine, 8—spermidine, 9—spermine.

**Figure 3 molecules-27-08194-f003:**
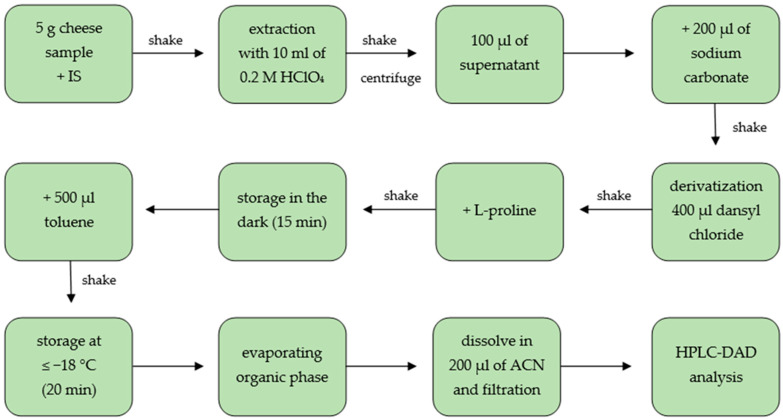
Scheme of sample preparation for the determination of BAs in cheese (IS: internal standard, ACN: acetonitrile, HPLC-DAD: high performance liquid chromatography with diode array detector).

**Table 1 molecules-27-08194-t001:** Results of the optimization on the number of extractions.

Biogenic Amines	Recovery (%)	
	Single Extraction	Double Extraction
Tryptamine	94.6	97.9
2-phenylethylamine	106.3	106.6
Putrescine	83.8	79.0
Cadaverine	97.6	91.9
Histamine	109.4	103.1
Tyramine	99.8	98.4
Spermidine	75.6	68.9
Spermine	75.5	66.7

**Table 2 molecules-27-08194-t002:** Comparison of biogenic amine peak areas of standard solution (25 µg/mL) derivatized by dansyl chloride in four different temperature-time conditions.

Biogenic Amine	Derivatization Conditions			
	25 °C for 15 min	40 °C for 45 min	60 °C for 5 min	60 °C for 15 min
Tryptamine	20.14	19.41	19.02	19.80
2-phenylethylamine	26.52	26.28	26.36	27.07
Putrescine	66.80	68.77	68.44	71.25
Cadaverine	58.74	58.89	58.45	60.73
Histamine	47.94	48.13	47.29	51.05
Tyramine	44.47	43.60	43.35	45.12
Spermidine	52.78	55.94	54.20	57.71
Spermine	42.19	47.23	44.89	48.51

**Table 3 molecules-27-08194-t003:** Parameters of calibration curves for quantification of biogenic amines and LOD and LOQ values.

Biogenic Amines	LOD (mg/kg)	LOQ (mg/kg)	Regression Equation, Y	Linearity	R^2^
Tryptamine	1.55	5.18	0.0182x + 0.0045	5.18-200	0.9998
2-phenylethylamine	1.59	5.31	0.0229x + 0.0056	5.31-200	0.9998
Putrescine	1.72	5.73	0.0591x + 0.0348	5.73-200	0.9997
Cadaverine	1.77	5.90	0.0513x + 0.0160	5.90-200	0.9997
Histamine	1.53	5.10	0.0425x + 0.0008	5.10-200	0.9998
Tyramine	1.60	5.33	0.0383x + 0.0092	5.33-200	0.9998
Spermidine	1.88	6.28	0.0497x + 0.0248	6.28-200	0.9997
Spermine	1.54	5.13	0.0432x + 0.0011	5.13-200	0.9998

**Table 4 molecules-27-08194-t004:** Accuracy of biogenic amines determination in cheese at three concentration levels: L1—50 mg/kg, L2—100 mg/kg, and L3—200 mg/kg.

Biogenic Amine	Intra-Day Precision (% RSD) (*n* = 6)			Inter-Day Precision (% RSD) (*n* = 18)			Recovery (% R) (*n* = 18)		
	L1	L2	L3	L1	L2	L3	L1	L2	L3
Tryptamine	2.5	2.3	2.2	11.8	7.2	10.5	80.7	77.3	78.1
2-phenylethylamine	2.9	2.0	2.4	8.0	6.7	7.0	98.0	91.8	93.3
Putrescine	2.6	3.0	1.1	10.5	4.3	5.9	96.1	89.0	91.3
Cadaverine	1.8	2.1	1.5	8.5	4.1	5.7	100.2	93.6	95.1
Histamine	2.1	1.4	2.0	12.4	8.1	7.7	94.2	93.0	91.0
Tyramine	1.7	0.7	2.2	9.3	12.2	13.1	85.2	82.9	80.8
Spermidine	2.6	2.3	0.7	9.7	8.9	6.4	81.0	80.3	80.7
Spermine	1.7	2.4	1.0	15.6	12.6	13.5	75.8	76.1	74.5

**Table 5 molecules-27-08194-t005:** The content of biogenic amines (TRYP—tryptamine, PHE—2-phenylethylamine, PUT—putrescine, CAD—cadaverine, HIS—histamine, TYR—tyramine, SPD—spermidine, SPM—spermine) determined in different ripened cheese types.

Samples	Biogenic Amines Concentrations (mg/kg)								
	TRYP	PHE	PUT	CAD	HIS	TYR	SPD	SPM	Total
Mould-ripened soft cheeses (*n* = 15)									
Brie	-	-	-	-	-	-	-	-	-
Brie	9.84	-	-	-	-	-	9.82	-	19.7
Camembert	-	-	-	-	-	-	7.45	-	7.45
Camembert	6.46	-	-	-	-	-	-	-	6.46
Camembert	-	-	-	-	-	-	6.68	-	6.68
Camembert	-	-	22.9	162	-	-	-	-	185
Caprice des dieux	-	-	-	-	-	-	9.86	-	9.86
Chruśniak z kozieradką	-	-	40.1	10.1	13.1	102	-	-	165
Limburger	14.1	-	-	-	6.48	19.0	-	-	39.6
Limburger	-	-	-	-	-	-	-	-	-
Saint Albray	-	-	-	-	-	-	-	-	-
Savaron	-	-	-	81.3	-	-	-	-	81.3
Snack a la francaise	-	-	-	-	-	-	-	-	-
Soft cheese	-	-	20.6	-	-	-	-	-	20.6
Soft cheese with chives	-	-	31.9	208	-	692	-	-	932
Blue-veined chesses (*n* = 10)									
Blu-fou	-	-	-	-	-	-	7.97	-	7.97
Bleu onctueux	7.99	6.91	6.57	61.0	-	-	15.1	-	97.6
Blu rival	-	-	-	-	-	-	13.4	-	13.4
Dorblu	-	-	-	-	-	-	-	-	-
Gorgonzola	-	-	-	-	94.2	-	7.05	-	101
Gorgonzola	-	7.39	-	-	127	-	-	-	134
Gorgonzola	-	-	-	-	78.4	36.6	11.2	-	126
Roquefort	-	15.1	7.46	59.7	-	631	-	-	713
Saint Agur	-	-	7.97	-	-	-	10.3	-	18.3
Srebrzysty	-	-	-	-	-	-	14.5	-	14.5
Hard cheeses (*n* = 10)									
Corregio	-	-	-	-	-	7.28	-	-	7.28
Emmental	-	42.5	15.4	14.4	15.8	474	-	-	562
Grana Padano	-	-	-	-	84.7	-	-	-	84.7
Grana Padano	-	-	-	-	11.5	-	-	-	11.5
Goat caciotta type	-	-	24.9	10.5	-	53.0	-	-	88.4
Goat caciotta type with pistachios	-	-	-	-	-	34.3	-	-	34.3
Le Gruyere	-	-	-	55.8	-	-	-	-	55.8
Parmigiano Reggiano	-	-	-	-	5.81	-	-	-	5.81
Raclette	-	27.3	170	148	-	287	-	-	632
Santtum	-	-	-	-	-	-	7.31	-	7.31

**Table 6 molecules-27-08194-t006:** Statistical evaluation of biogenic amines concentration within three cheese type groups.

Samples	Biogenic Amines Concentrations (mg/kg)								
	TRYP	PHE	PUT	CAD	HIS	TYR	SPD	SPM	Total
Mould-ripened soft cheeses (*n* = 15)									
Minimum	6.46	-	20.6	10.1	6.48	19.0	6.68	-	6.46
Mean	10.3	-	28.9	115	9.79	271	8.45	-	73.7
Median	10.3	-	27.4	122	9.79	102	8.64	-	16.6
90 Percentile	13.3	-	37.6	194	12.4	574	9.85	-	167
95 Percentile	13.7	-	38.9	201	12.8	633	9.85	-	232
Maximum	14.1	-	40.1	208	13.1	692	9.86	-	692
Blue-veined chesses (*n* = 10)									
Minimum	7.99	6.91	6.57	59.7	78.4	36.6	7.05	-	6.57
Mean	-	9.80	7.33	60.4	99.9	334	11.4	-	58.4
Median	-	7.39	7.46	60.4	94.2	334	11.2	-	13.4
90 Percentile	-	13.6	7.87	60.9	120	572	14.7	-	94.2
95 Percentile	-	14.3	7.92	60.9	124	601	14.9	-	127
Maximum	-	15.1	7.97	61.0	127	631	15.1	-	631
Hard cheeses (*n* = 10)									
Minimum	-	27.3	15.4	10.5	5.81	7.28	7.31	-	5.81
Mean	-	34.9	70.1	57.2	29.5	171	-	-	78.4
Median	-	34.9	24.9	35.1	13.7	53.0	-	-	27.3
90 Percentile	-	41.0	141	120	64.0	399	-	-	193
95 Percentile	-	41.7	156	134	74.4	437	-	-	305
Maximum	-	42.5	170	148	84.7	474	-	-	474

## Data Availability

Not applicable.

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
