# Peer review of "Development of a High Performance Liquid Chromatography with Diode Array Detector (HPLC-DAD) Method for Determination of Biogenic Amines in Ripened Cheeses"

_molecules, 2022, doi:10.3390/molecules27238194_

Round 1

Reviewer 1 Report

The manuscript is well prepared and could be considered for acceptance after some modifications.

1. Dansyl chloride is a well-known fluorescence labelling reagent. The diode array detector was used as the detector. Why didn't the authors choose fluorescence detector with better sensitivity and selectivity?

2. Chemical derivatization is a straightforward and powerful tool for analyzing analytes of interests in complex samples. In introduction part, the authors should add some discussions about chemical derivatization to help the readers understand the advantages of derivatization. Some useful articles were listed as follows:

(1) A simultaneous extraction/derivatization strategy coupled with liquid chromatography–tandem mass spectrometry for the determination of free catecholamines in biological fluids, Journal of Chromatography A, 2021, 1654, 462474.

(2) Stable isotope labelling-flow injection analysis-mass spectrometry for rapid and high throughput quantitative analysis of 5-hydroxymethylfurfural in drinks, Food Control 2021, 130, 108386.

(3) High throughput and very specific screening of anabolic-androgenic steroid adulterants in healthy foods based on stable isotope labelling and flow injection analysis-tandem mass spectrometry with simultaneous monitoring proton adduct ions and chloride adduct ions,Journal of Chromatography A, 2022,1667,462891.

3. Molecules is an open-access journal. There’s no extra cost for color figures. I suggest the authors use color figures.

4. A schematic diagram of the whole process would be helpful for the readers to understand the article.

5. For figure 1, error bar should be given.

Reviewer 2 Report

The presented manuscript shows the validation of the determination method of biogenic amines in cheese. The work seems good and adequate for publication in the journal. just the statistical analysis section missed; please add a specific data analysis method.

Round 2

Reviewer 1 Report

The revised manuscript could be considered for acceptance.